# Response of Grain Yield to Planting Density and Maize Hybrid Selection in High Latitude China—A Multisource Data Analysis

Shanwen Sun [1], Zhaofu Huang [2,*], Haiyan Liu [1], Jian Xu [1], Xu Zheng [1,3], Jun Xue [2] and Shaokun Li [2]

[1]  Qiqihar Branch of Heilongjiang Academy of Agricultural Science, Qiqihar 161006, China; yzjcat250@163.com (S.S.); ztlovezx@163.com (X.Z.)
[2]  Chinese Academy of Agricultural Sciences/Key Laboratory of Crop Physiology and Ecology, Institute of Crop Sciences, Ministry of Agriculture and Rural Affairs, Beijing 100081, China; lishaokun@caas.cn (S.L.)
[3]  Engineering Research Center of Agricultural Microbiology Technology, Ministry of Education, Heilongjiang University, Harbin 150500, China
*   Correspondence: 17610498208@163.com; Tel.: +86-010-82108891

**Abstract:** Identifying the relationships between the yield of rainfed maize and planting densities as well as the hybrids used is crucial for ensuring the sustainable development of the grain industry in high latitude China. In this study, we collected 108 grain yield date points from our multiyear (2017–2020) field experiments and combined 213 data points collected from 21 published papers to appraise the impact of planting density and hybrids on maize yield. It was found that grain yield forms a curvilinear relationship with plant density as it increased from 22,500 to 112,500 plants ha$^{-1}$. The optimum plant density (OPD) was determined to be 72500 plants ha$^{-1}$, with a maximum maize grain yield of 10.56 Mg ha$^{-1}$. The interannual variability in grain yields among hybrids with different planting densities was mainly due to the differences in dry matter (DM), especially post-silking. Grain yields increased significantly with a rise in the proportion of post-silking DM to DM at maturity. In addition, both the collected literature and our field experiments showed that the OPD was positively correlated with solar radiation accumulated during the maize growth period and with each hybrid's year of release. This study suggests that increasing plant density and selecting new hybrids with suitable growth periods are effective approaches for increasing grain yield in high latitude China.

**Keywords:** plant density; grain yield; dry matter; high latitude China

## 1. Introduction

Since 2013, the rise of maize as an important feed for cattle and sheep has made the crop China's largest food source, with the expanding Chinese production now accounting for 21% of global maize production [1,2]. Despite this increase in popularity and production, the world is likely to face food shortages in the coming decades as the global population increases [3,4]. Therefore, it is essential to maximize grain yield from maize production areas to ensure food security [5]. From the 1960s onward, the genetic improvements in hybrids and the increase in planting densities have contributed to a rise in global maize production by 168.5 and 311.8%, respectively [6]. Many growers have found that higher density planting was the most effective method of increasing grain yield [7–9]. Previous studies have indicated that the maize yield in China can increase by 2.75–10.5% when the maize planting density reaches 15,000 plants ha$^{-1}$, without excess nitrogen fertilization [4]. Increasing the planting density can also improve the photosynthetic utilization efficiency, and the accumulation of dry matter [10–12].

Up until 2016, the average planting density was 59,100 plants ha$^{-1}$ in high latitude China—significantly lower than the average United States planting density of 82,500 plants ha$^{-1}$ [5,13]. The average maize grain yield is also much lower, with only

4.7 Mg ha$^{-1}$ compared to 11.09 Mg ha$^{-1}$ in the U.S. [14]. Chinese hybrids released in the 1990s had a low optimum planting density, with an estimated 65,000 plants ha$^{-1}$ compared to 85,000 plants ha$^{-1}$ for Brazilian hybrids released during the same time period. Over the past 50 years, U.S. maize planting densities have increased by an average rate of approximately 1000 plants ha$-1$ per year while they have only increased in high latitude China by 175 ha$^{-1}$ per year [6,15]. Several previous studies have shown that intercepted solar radiation during the crop growth period significantly impacts the optimum maize plant density [16,17]. In addition to having a higher ability to absorb light, modern hybrids are more tolerant to prolonged periods of solar radiation. Climate factors such as excessive rainfall can significantly alter available sunlight [18], and they result in deviations in optimal plant density and crucial grain yields [19].

High latitude China is a major agricultural production area, with maize accounting for more than 13.3% of the nation's total crop sowing areas and 14.0% of total grain production. However, the planting density is generally low, and grain yield varies greatly between years in high latitude China. It is unclear how yield responds to changes in planting density under field conditions. Unfortunately, this lack of understanding has impeded progress in the development of new breeding hybrids and innovative planting technologies. We have conducted over four years of field experiments and have integrated previous research in order to elucidate the correlation between maize planting density and yield, as well as the relationship between cumulative solar radiation and grain yield. Through this, we have identified the optimal planting density for high latitude China, providing a foundation to support the sustainable development of maize production. This research offers valuable theoretical support for the advancement of maize production in high latitude China.

## 2. Materials and Methods

### 2.1. Site Description and Weather Data

During the 2017–2020 maize growing seasons, field experiments were conducted at research stations in Qiqihar (Heilongjiang, China, 47°36′ N, 123°92′ E), Suihua (Heilongjiang, China, 46°66′ N, 126°98′ E), and Jiamusi (Heilongjiang, China, 46°81′ N, 130°33′ E). The location of each site is shown in Figure 1. The average temperatures in Qiqihar, Suihua, and Jiamusi were 19.7, 19.1, and 18.6 °C, respectively, and the annual rainfall was 469.0, 576.7, and 628.8 mm, respectively (Table 1). Soil testing was not completed during the period of the experiment; however, from the literature data of 2014, the elements in the soil were as follows: 33.7 g kg$^{-1}$ organic matter, 1.5g kg$^{-1}$ total N, 45.6 mg kg$^{-1}$ of available P, and 226.6 mg kg$^{-1}$ of available K in the top 0–40 cm arable soil layer [20].

### 2.2. Experimental Design and Field Management

This study used three typical commercial hybrids Nendan23 (ND23), Suiyu23 (SY23), and Heyu29 (HY29) in high latitude China, obtained from the Qiqihar Branch of Heilongjiang Academy of Agricultural Science, Suihua Branch of Heilongjiang Academy of Agricultural Science, and Jiamusi Branch of Heilongjiang Academy of Agricultural Science. The planting area of the three hybrids exceeds one million hectares in high latitude China, and the details of each hybrid are listed in Table 2. Each trial utilized a split-plot randomized complete block design with three replicates. Each hybrid was planted at three densities, 60,000 (D1), 67,500 (D2), and 75,000 (D3) plants ha$^{-1}$. The trials were established once per year, with maize sown by hand in late April and early May and harvested in late September and early October. Controlled-release fertilizer was applied at each site before sowing, and included 56 kg N ha$^{-1}$, 41 kg P ha$^{-1}$, and 60 kg K ha$^{-1}$. To prevent weeds from competing with the maize, refined metolachlor was applied prior to emergence. At the 11–12th leaf stage, a drone was used to spray difenoconazole in order to protect the maize against pests and diseases.

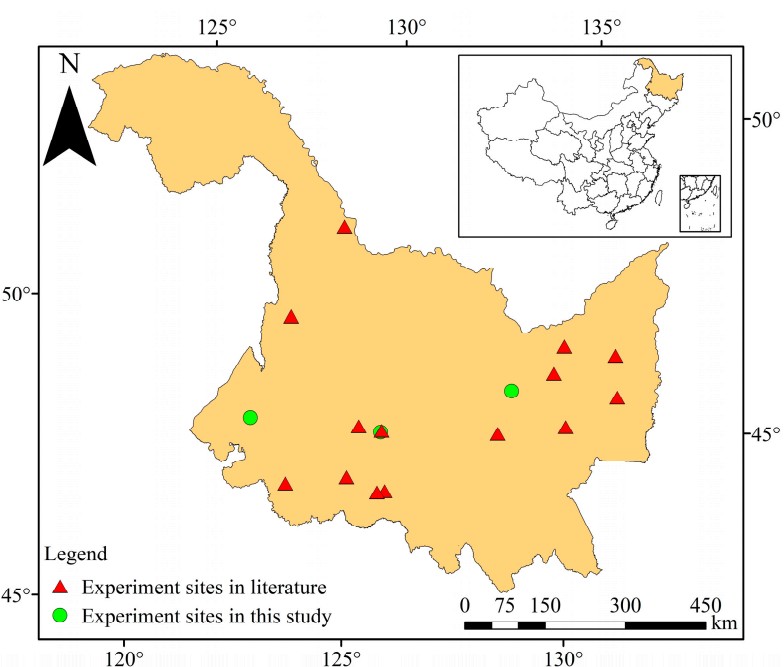

**Figure 1.** Distribution of experiment sites in the collected literature (red triangles) and this study (green circles).

**Table 1.** Maize growing season (May–October) rainfall and mean daily air temperature in 2017, 2018, 2019, and 2020 at study sites.

| Experimental Site | Precipitation (mm) | | | | Mean Daily Air Temperature (°C) | | | |
|---|---|---|---|---|---|---|---|---|
| | 2017 | 2018 | 2019 | 2020 | 2017 | 2018 | 2019 | 2020 |
| Qiqihar | 247.0 | 557.5 | 525.0 | 547.0 | 20.0 | 20.0 | 19.2 | 19.6 |
| Suihua | 388.4 | 442.0 | 778.0 | 698.0 | 19.2 | 19.4 | 18.6 | 19.1 |
| Jiamusi | 481.2 | 627.6 | 829.0 | 577.0 | 18.9 | 18.5 | 18.1 | 18.8 |

**Table 2.** Features of maize hybrids used in the experiments.

| Hybrid | Breeding Institute | Year of Release | Relative Maturity |
|---|---|---|---|
| Nendan23 (ND23) | Qiqihar Branch of Heilongjiang Academy of Agricultural Science | 2019 | 122 |
| Suiyu23 (SY23) | Suihua Branch of Heilongjiang Academy of Agricultural Science | 2011 | 120 |
| Heyu29 (HY29) | Jiamusi Branch of Heilongjiang Academy of Agricultural Science | 2017 | 125 |

*2.3. Sampling and Measurements*

Maize growth stages are defined as the budding stage (VE), 6th leaf stage (V6), 12th leaf stage (V12), tasseling stage (VT), silking stage (R1), lactation stage (R3), mass stage (R4), and physiological maturity stage (R6). Data were collected when at least 50% of the plants reached each growth stage. Three uniform maize plants were harvested from the middle row of each plot at silking and maturity. These plants were divided into stems (including tassels), leaves, sheaths, cobs, bracts, and kernels, and all isolated components were oven dried at 85 °C for 48 h (Hongjing, China). The weights were recorded once they became constant. DM accumulation was calculated as follows: DM = DM per plant × density; DM at pre-silking + DM at post-silking = DM at maturity. At maturity, the ears from 66.7 m$^2$

area in the middle of each plot were manually harvested and the grain moisture content was gauged using a PM-8188 portable grain moisture meter (Kett, Tokyo, Japan), and the ultimate grain yield was measured at a standardized level of 14.0% moisture.

### 2.4. Data Collection from Previous Studies

Relevant papers published between 1990 and 2021 using the search terms "maize or corn hybrids", "density", and "China" were obtained from Web of Science, Springer Link, the China Knowledge Resource Integrated Database, and Scikit. The standards used for including publications in our analysis were as follows: (i) validity of grain yield (i.e., could be standardized at 14% grain moisture content and two or more plant density levels); (ii) research was conducted between 1990 and 2021; (iii) location of study was within high latitude China. Most of the data were retrieved from tables, and the remainder were taken from figures digitized using the WebPlotDigitizer software. We collected 231 grain yield date points from 21 publications, respectively. The experiment locations of the literature studies are shown in Figure 1.

### 2.5. Statistical Analysis

Statistical analysis was performed using IBM SPSS 26 software (IBM Corporation, Armonk, NY, USA). Analysis of variance was conducted using year, hybrid, planting density, and site, as well as their effects on dry matter and grain yield. Correlation analysis was used to determine the relationship between various indices and grain yield. The figures were created using ArcMap14.0, Origin 2021 (OriginLab, Northampton, MA, USA), SigmaPlot 14.0 (Systat Software, San Jose, CA, USA).

## 3. Results

### 3.1. Maize Yield and Yield Stability Analysis

From 2017–2020, the total average grain yields of Qiqihar, Suihua, and Jiamusi were 10.27 Mg ha$^{-1}$, 11.89 Mg ha$^{-1}$, and 11.12 Mg ha$^{-1}$, respectively. Different plant densities of Qiqihar and Suihua inconsistently resulted in differences in grain yield. Conversely, Jiamusi showed no significant difference. The total average grain yield of HY29 was significantly higher than that of SY23 and ND23 in both Suihua and Jiamusi. There was no significant difference between the three hybrids in Qiqihar (Figure 2). The intra-annual coefficient of variation (CV) of maize yield ranged from 3.24 to 22.42%, with an average of 10.63%. The annual average CV of maize yield with densities 60,000, 67,500, and 75,000 plants ha$^{-1}$ was 9.53, 12.16 and 10.26%, respectively. The highest CV (16.43%) was observed in Qiqihar, while the lowest (6.21%) was in Jiamusi. In addition, the intra-annual CV of HY29 was lower than SY23 and ND23 (Figure 3). As shown in Table 3, yield, ear density, and thousand-kernel weight were significantly affected by the interaction of H × D; Y × H, Y × D, and Y × H × D were not significant for yield.

**Table 3.** ANOVA analysis for the effects of year, hybrid, and planting density on the grain yields, and yield components.

| Source | Yield | Ear Density | Kernel Number per Ear | 1000-Kernel Weight |
|:---:|:---:|:---:|:---:|:---:|
| Year (Y) | ** | ** | ** | ** |
| Hybrid (H) | ** | ** | ns | ** |
| Density (D) | * | ** | ** | ** |
| Y × H | ns | ** | ns | ** |
| Y × D | ns | ns | ns | ** |
| H × D | * | * | ns | ** |
| Y × H × D | ns | * | ** | ** |

* and ** indicate significant differences at $p < 0.05$ and $p < 0.01$ probability levels, and ns indicates no significance, respectively.

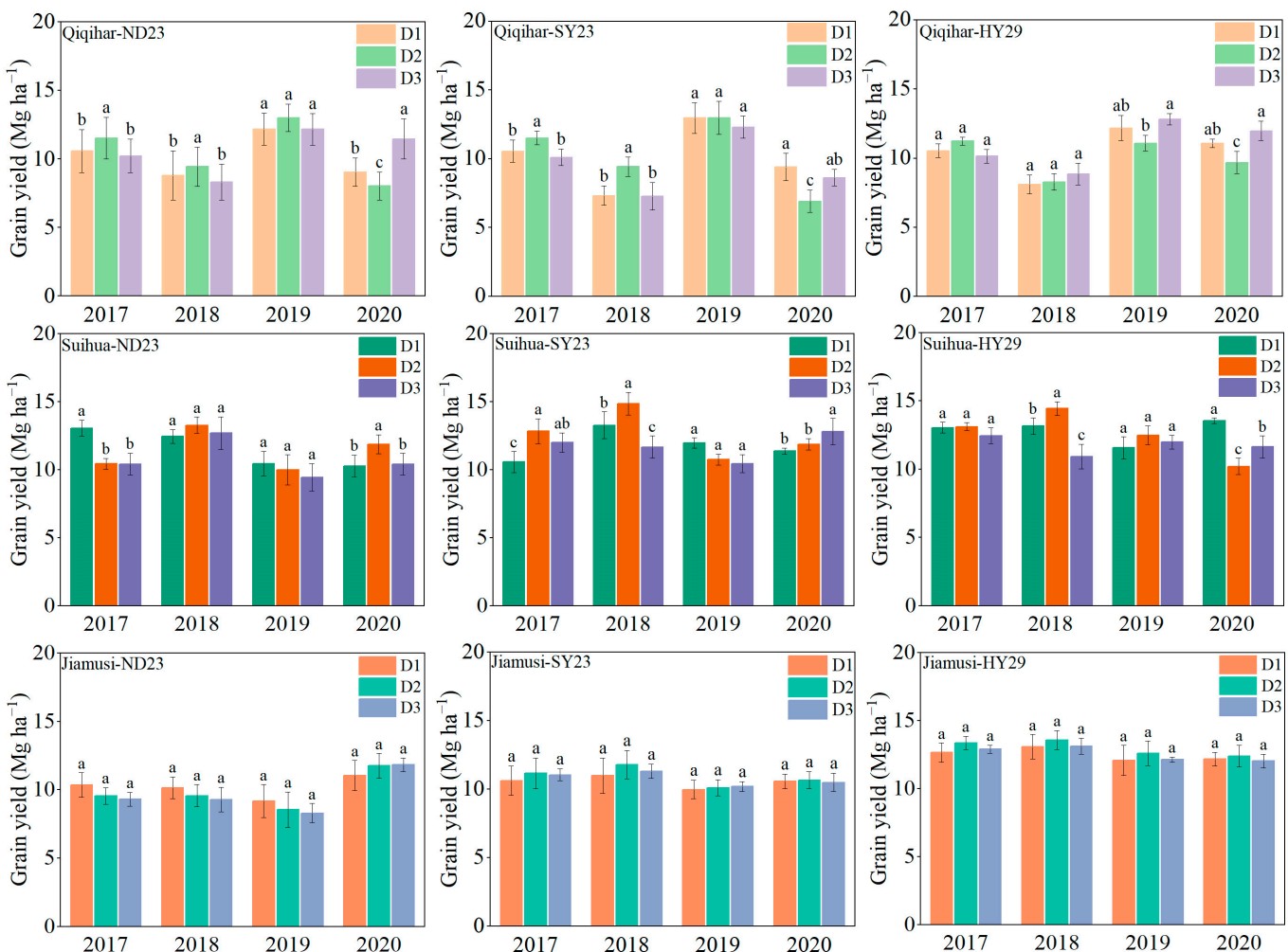

**Figure 2.** Grain yield between different densities in Qiqihar, Suihua, and Jiamusi for ND23, SY23, and HY29 from 2017 to 2020. Values followed by the same letter were not significantly different at $p < 0.05$ according to Fisher's LSD significant difference test. Error bars represent ± standard error of the mean.

### 3.2. Dry Matter Accumulation

The DM accumulation of the population at the silking stage and maturity stage increased with the rise in planting densities, and the interactions between density and site and hybrid and site significantly affected dry matter accumulation (Table 4). Across growth stages, the DMs of 75,000 plants ha$^{-1}$ and 67,500 plants ha$^{-1}$ density plantings were higher than those of 60,000 plants ha$^{-1}$. The DM accumulation of the population in Qiqihar was significantly higher than that in Suihua and Jiamusi. Further analysis showed no significant correlation between grain yield and DM at silking (Figure 4a), but a significant positive correlation at post-silking and maturity (Figure 4b,c). Grain yield increased significantly with a decrease in the ratio of DM to total DM before silking, and with the increase in the ratio of DM to total DM after silking (Figure 4d). The average proportion of post-silking DM (63.4%) was significantly higher than pre-silking DM (36.2%).

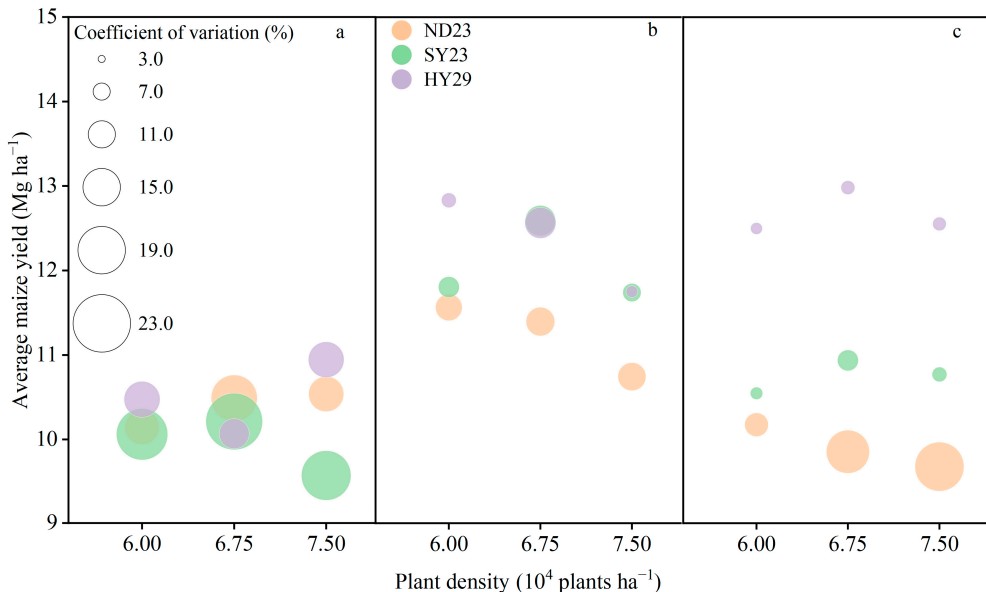

**Figure 3.** Coefficient of variation of maize yield under different densities in Qiqihar (**a**), Suihua (**b**), and Jiamusi (**c**) from 2017 to 2020.

**Table 4.** Influence of year, hybrid, planting density, and test site on population dry matter accumulation.

| Source of Variation | | Population Dry Matter Accumulation (Mg ha$^{-1}$) | | |
|---|---|---|---|---|
| | | Silking | Post-Silking | Maturity |
| Density (plants ha$^{-1}$) | 60,000 | 10.19 ± 1.77 a | 11.18 ± 2.88 ab | 27.82 ± 3.49 c |
| | 67,500 | 10.29± 1.73 a | 11.52 ± 3.28 a | 30.93 ± 4.15 b |
| | 75,000 | 10.98 ± 1.94 a | 10.81 ± 3.30 b | 31.73 ± 4.24 a |
| Hybrid | ND23 | 9.33 ± 1.25 b | 10.47 ± 2.62 b | 29.34 ± 3.07 b |
| | SY23 | 10.33 ± 1.88 ab | 11.17 ± 3.08 ab | 29.52 ± 4.37 b |
| | HY29 | 10.80 ± 1.86 a | 11.92 ± 3.35 a | 31.61 ± 4.41 a |
| Site | Qiqihar | 11.91 ± 1.51 a | 10.73 ± 3.95 a | 33.05 ± 4.43 a |
| | Suihua | 9.50 ± 1.36 b | 11.18 ± 2.90 a | 27.84 ± 3.58 c |
| | Jiamusi | 10.05 ± 1.25 b | 11.21 ± 2.62 a | 29.59 ± 3.07 b |
| Year | 2018 | 9.83 ± 1.73 a | 11.31 ± 3.14 a | 30.54 ± 4.27 a |
| | 2019 | 10.39 ± 1.52 a | 10.95 ± 3.46 a | 30.04 ± 3.61 a |
| | 2020 | 10.23 ± 2.45 a | 11.26 ± 3.19 a | 29.89 ± 4.94 a |
| Density (D) | | ns | ns | ** |
| Hybrid (H) | | * | * | * |
| Site (S) | | * | * | ** |
| Year (Y) | | ns | ns | ns |
| D × H | | ns | ns | ** |
| D × S | | * | * | ** |
| H × S | | ** | ** | ** |
| H× Y | | * | * | * |
| S × Y | | ns | ns | * |
| D × H × S | | * | * | * |
| D × H × Y | | ns | ns | * |
| S × H × Y | | * | * | * |
| D× H × S × Y | | * | ** | ** |

*, significant at $p < 0.05$. **, significant at $p < 0.01$. ns, not significant. The values are the means ± standard deviations. Values followed by the same letter are not significantly different at $p < 0.05$ according to Duncan's significant difference test.

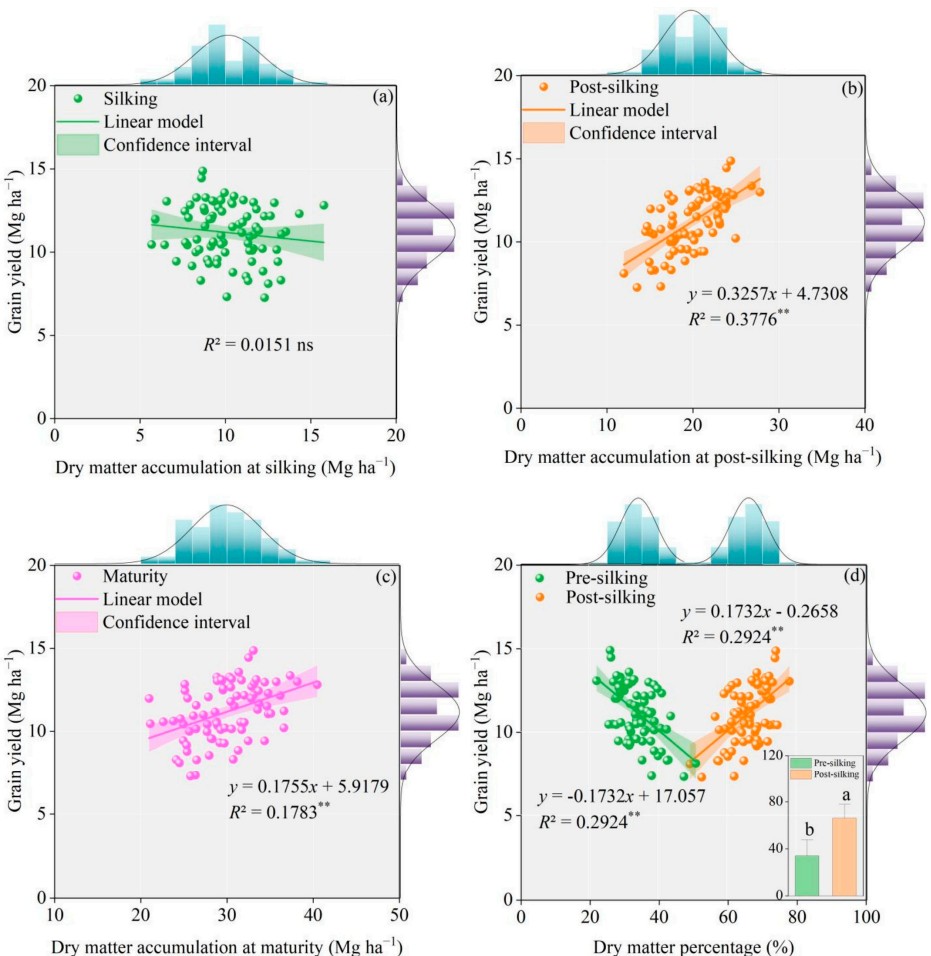

**Figure 4.** Effects of DM accumulation at silking (**a**), post-silking (**b**), and maturity (**c**), and the proportion of pre-silking and post-silking DM (**d**) on grain yield. \*\*, significant correlation at the level of $p < 0.01$. ns, not significant.

### 3.3. Response of Grain Yield to Plant Density

For plant density analysis, plots ranged from 22,500 to 112,500 plants ha$^{-1}$ and were normally distributed with an average of 67,300 plants ha$^{-1}$ (Figure 5a). The relationship between plant densities and grain yields was found to be a quadratic curve with an estimated optimum plant density of 72,500 plants ha$^{-1}$ and a corresponding grain yield of 10.56 Mg ha$^{-1}$. Our analysis revealed that grain yields increased linearly with the year of the hybrid's release. Hybrids in the 2010s were found to produce significantly higher yields than varieties in the 1970s to 2000s (Figure 5b). Therefore, the newer hybrids showed clear increases in tolerance of dense planting.

### 3.4. Relationship between Solar Radiation and Optimum Plant Density

The optimum plant density of maize was significantly positively correlated with cumulative solar radiation during the growth period based on data collected from previous studies (Figure 6a). This was consistent with our experimental finding where solar radiation was positively correlated with grain yield (Figure 6c). In addition, the optimal plant density increases significantly with the year of the maize hybrid's release (Figure 6b). The optimum grain yields (i.e., the maximum grain yields) of ND23, SY23, and HY29 were 11.29, 11.64, and 13.15 Mg ha$^{-1}$, respectively (Figure 6d).

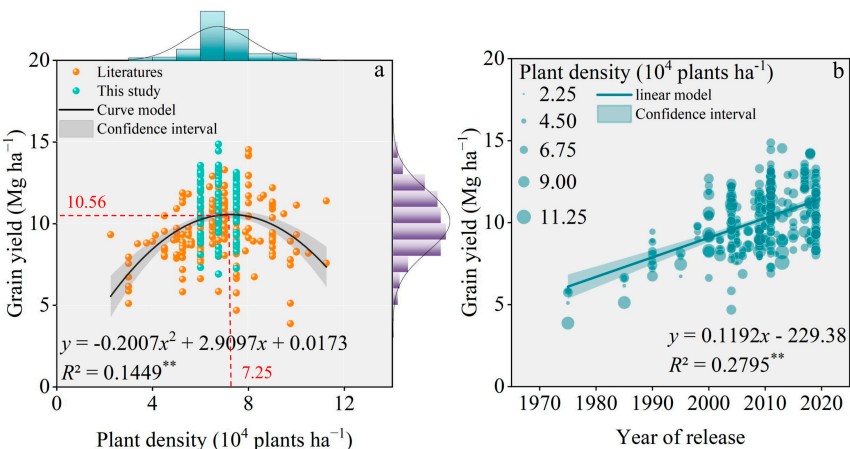

**Figure 5.** (**a**) Relationship between grain yields and density; (**b**) relationship between grain yields and year of maize hybrids' release. **, significant correlation at the level of $p < 0.01$.

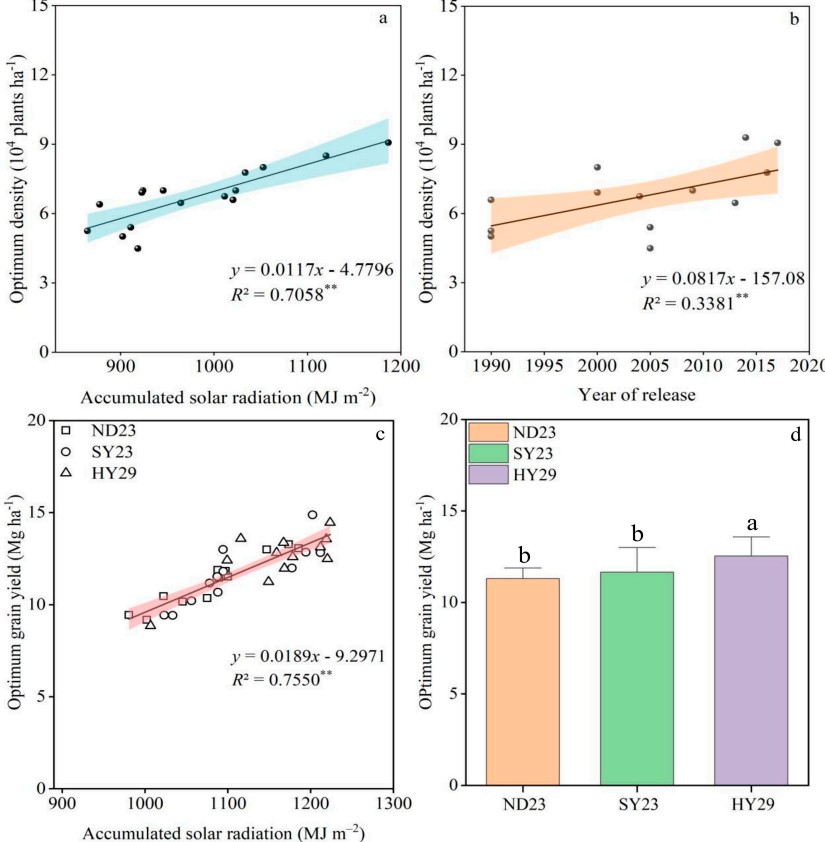

**Figure 6.** (**a**) The relationship between optimum plant density and the accumulated solar radiation of the growth period (n = 16). The black line and blue area represent the linear model and confidence interval, respectively. (**b**) The relationship between optimum plant density and year of maize hybrids' release in the collected literature. The black line and yellow area represent the linear model and confidence interval, respectively. (**c**) Relationship between hybrids' grain yield of optimum density and the accumulated solar radiation of the growth period from our field experiments. The black line and red area represent the linear model and confidence interval, respectively. (**d**) The optimum grain yield of tested maize hybrids. Different lowercase letters above the bars indicate significant difference among tested hybrids ($p < 0.05$). Error bars represent ± standard error of the mean. **, significant at the level of $p < 0.01$.

## 4. Discussion

### 4.1. Effect of Dry Matter Accumulation on Grain Yield

Many previous studies have proven that DM is the basis for maize grain yield formation [21,22], with increases in DM often attributed to larger grain yield [23–25]. Interannual variations of grain yield for the same hybrid have been linked to the change in DM [26]. Data collected from our four years of field experiments showed that the DM content of post-silking and maturity significantly increased with grain yield (Figure 4). However, a recent study found no significant increase in DM before silking when grain yield surpassed 17.0 Mg ha$^{-1}$ [27]. This suggests that pre-silking DM is essential for grain yield in dense plantings, similar to our results (although we did not reach a 17.0 Mg ha$^{-1}$ production level) (Figure 4). There is much evidence that DM before silking is crucial for modern high-yielding maize hybrids [28–30], and can increase by up to 70% when grain yield increases beyond 18 Mg ha$^{-1}$ [31]. Similarly, we found that as with grain yield, the proportion of pre-silking DM decreased significantly, while that of post-silking DM increased significantly. The average proportions of pre-silking DM and post-silking DM were 36.2 and 63.4%, respectively (Figure 4d), which are consistent with the previous study [27]. This is because more than 50% of the total dry matter of the maize plant is obtained at silking, with most of this post-silking DM occurring in the grain. Therefore, the post-silking dry matter plays a decisive role in the variation in maize grain yield [32,33].

### 4.2. Relationship between Solar Radiation and Planting Density and Yield

Solar radiation plays a significant role in maize production. Not only does it increase photosynthesis and crop yield, but it also supports the formation and development of plant organs [34,35]. Some studies reported that global effective solar radiation has been reduced by an average of 1.4–2.7% per decade, leading to an estimated 3–6% loss of maize yield in the Huanghuaihai Plain of China [36]. The same studies also demonstrate that the yield is significantly improved when solar radiation matches the optimal plant density [19]. Consistent with previous studies [37], our analysis shows that the accumulated solar radiation was significantly associated with optimal maize plant density based on data collected from previous studies (Figure 6a). Previous research shows that the optimum plant density is 85,500 plants ha$^{-1}$ in high latitude China [38], which is significantly higher than our result of 72,500 plants ha$^{-1}$ (Figure 5a). In addition, some studies have shown differences in the quantity of photosynthetic products transferred to the grain under low radiation among different hybrids, and therefore differences in maize yield [39]. In this study, the grain yield of HY29 under the optimum density was significantly higher than that of ND23 and SY23. Therefore, selecting suitable hybrids may reduce the yield loss caused by a lack of light radiation [40].

### 4.3. Increasing Planting Density and Hybrid Selection to Increase Maize Yield in High Latitude China

Numerous researchers have reported that increasing DM accumulation by increasing plant density is a reliable way to obtain a high yield [41,42]. The relationship between grain yield and planting density can be a quadratic equation. According to the curve, the grain yield of maize increased with higher planting density and gradually decreased after reaching the maximum yield [43]. Recently, a study reported that the global optimal plant density of maize is between 90,000 and 120,000 plants ha$^{-1}$, and the corresponding highest grain yield ranges from 9.0 to 11.0 t ha$^{-1}$ [44]. By reviewing data from previous studies combined with data from four consecutive years of field experiments, we found that the optimum planting density was 72,500 plants ha$^{-1}$ for the location where the study was conducted and in the corresponding weather conditions, and the relative yield was 10.56 Mg ha$^{-1}$ (Figure 5a). This is significantly lower than the average planting density (85,000 plants ha$^{-1}$) of the U.S. maize belt at the same latitude in the 2010s [45]. Furthermore, other studies have suggested that newer hybrids have significantly higher grain yields per plant and per unit area than older hybrids, regardless of planting density [46,47].

Similar results found that grain yields increased with the increase in the hybrid's release year (Figure 6b), which can be attributed to genetic improvement [48,49]. Newer hybrids have also been found to be more tolerant to light stress [46]. Some studies have shown that the rate of barrenness and earless plants is higher in older hybrids than newer ones as planting density increases [50]. Newer hybrids also have shorter filament intervals, ensuring simultaneous rapid deposition and pollination [51,52]. The results show that there is still an extensive margin for improving maize yield and ensuring food security by increasing the planting density and selecting new hybrids with suitable growth periods in high latitude China.

## 5. Conclusions

Comprehensive analysis showed that the optimal planting density was 72,500 ha$^{-1}$, which was significantly affected by the accumulated solar radiation in high latitude China. In addition, the optimal planting density increased with the year of hybrid release. Differences in dry matter accumulation among hybrids were the main factor responsible for the annual variation in grain yield with different planting densities, especially post-silking. Hence, it is important to increase planting density, but farmers also need to consider selecting new hybrids with suitable growth periods to increase their maize grain yields in high latitude China.

**Author Contributions:** Conceptualization: Z.H. Experimental design: Z.H., J.X. (Jun Xue) and S.L. Methodology: S.S., Z.H., J.X. (Jun Xue) and S.L. Fieldwork: S.S., H.L., J.X. (Jian Xu) and X.Z. Data analysis: S.S. Writing—original draft: S.S. Funding acquisition: S.S. Writing—review and editing: Z.H. All authors have approved the version to be published. All authors have read and agreed to the published version of the manuscript.

**Funding:** This research was supported by grants from the China Agriculture Research System of MOF and MARA, and the Agricultural Science and Technology Innovation Program (CAAS-ZDRW202004).

**Institutional Review Board Statement:** Not applicable.

**Informed Consent Statement:** Not applicable.

**Data Availability Statement:** Not applicable.

**Conflicts of Interest:** The authors declare no conflict of interest.

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
