# Peer review of "Response of Grain Yield to Planting Density and Maize Hybrid Selection in High Latitude China—A Multisource Data Analysis"

_agronomy, doi:10.3390/agronomy13051333_

Round 1

Reviewer 1 Report (Previous Reviewer 2)

Dear Authors,

thank you for taking into account the corrections I have given regarding the manuscript. In my opinion, the manuscript has significantly improved in quality. 

As for the content of the work, I have two minor comments:

Since no soil testing was done for this experiment I believe that data from the literature should not be given. Alternatively, it can be written that, soil testing was not done during the period of the experiment while from the literature data of 20XX the elements in the soil were as follows. 

As it stands, the notation may be misleading, suggesting that soil testing was conducted.

Figure 2 I guess that the lines next to the bars of the figure indicate the standard deviation. Please add to in the explanation below the figure.

Table 4. Please add standard deviation values for all values.

sincerely

Author Response

Dear reviewers, Thanks very much for taking your time to review this manuscript. I really appreciate all your comments and suggestion! Please find my itemized responses in below and my revisions in the re-submitted files.Thanks again!

Comments and Suggestions for Authors

Dear Authors,thank you for taking into account the corrections I have given regarding the manuscript. In my opinion, the manuscript has significantly improved in quality.

As for the content of the work, I have two minor comments:

Since no soil testing was done for this experiment I believe that data from the literature should not be given. Alternatively, it can be written that, soil testing was not done during the period of the experiment while from the literature data of 20XX the elements in the soil were as follows. As it stands, the notation may be misleading, suggesting that soil testing was conducted.

Responses: We would like to thank you for giving us the opportunity to revise our manuscript. According to suggestions, , we have revised the details in lines 77 to 81 in the text.

Figure 2 I guess that the lines next to the bars of the figure indicate the standard deviation. Please add to in the explanation below the figure.

Responses: Thank you for comments. According to the referee’s suggestions, we have revised it. The detailed corrections are shown in line 153.

Table 4. Please add standard deviation values for all values.

Responses: Thank you for comments. According to the referee’s suggestions, we have revised it. The detailed corrections are shown in Table 4.

Reviewer 2 Report (New Reviewer)

Check the sticky notes

Improving the wording and use of units of measurement

Author Response

Dear reviewers, Thanks very much for taking your time to review this manuscript. I really appreciate all your comments and suggestion! Please find my itemized responses in below and my revisions in the re-submitted files.Thanks again!

Comments and Suggestions for Authors

Check the sticky notes

Responses: Dear reviewer, thanks very much for taking your time to review this manuscript. According to the referee’s suggestions, we have revised the details in the text. Thanks again!

Reviewer 3 Report (New Reviewer)

Reviewer:

This study analyzed many biologic-ecological parameters in three types of maize hybrids. Authors enriched their study with bibliographical results also carried out in China. I found that this study obtained relevant results, which will be interesting for scientific community as well as for general society. I consider that the study was carefully carried out and the statistical analysis were appropriate.

I only detected some typing errors, which were marked in color in the pdf of the article here attached.

In many parts of the text and in legends of figures a dot not needed was written after the word Figure. #.

In line 106: eliminate “,” after the word the

In line 121: ii) one of the two words “was” is not necessary.

In line 165: the word plants is wrong written after: .... 6000 plnats .... .

In line 109: clarify what do you mean with the sentence “weights were recorded once they become constant”, a brief explanation to clarify this sentence is necessary.

I consider that in the figures the conventional acronym of Mg (in capitalized letter M) was misused since it is assigned to magnesium. It seems that in true the legend of vertical axis should be say milligram; whose acronym is with lower letters: mg. Thus I recommend to use mg instead Mg https://www.collinsdictionary.com/dictionary/english/mg#:~:text=mg%20is%20a%20written%20abbreviation%20for%20milligram%20or%20%2C%20milligrams.

Figure 4a) Is r or R? Moreover, I suggest to include a space between the number and the ns word.

I could not find the figure 5, after figure 4 was presented figure 6, by the way, this last figure was referred in line 181 but with a dot that is not necessary (Figure. 6a).

In Line 183, was presented an interesting result .....plant density of 72,500 plants ha-1: Clarify, if this interesting result is that showed in the Figure 6a? If, it is, please refer the figure in this sentence.

In the figure 7, it was confused that the ordinate has a negative value. Please give a brief explanation about this negative value.

Author Response

Dear reviewers, Thanks very much for taking your time to review this manuscript. I really appreciate all your comments and suggestion! Please find my itemized responses in below and my revisions in the re-submitted files.Thanks again!

Comments and Suggestions for Authors

This study analyzed many biologic-ecological parameters in three types of maize hybrids. Authors enriched their study with bibliographical results also carried out in China. I found that this study obtained relevant results, which will be interesting for scientific community as well as for general society. I consider that the study was carefully carried out and the statistical analysis were appropriate. I only detected some typing errors, which were marked in color in the pdf of the article here attached. In many parts of the text and in legends of figures a dot not needed was written after the word Figure.

In line 106: eliminate “,” after the word the

Responses: Thank you very much. According to the referee’s suggestions, we have revised the details in lines 107 in the text.

In line 121: ii) one of the two words “was” is not necessary.

Responses: Thank you very much. According to the referee’s suggestions, we have revised the details in lines 122 in the text.

In line 165: the word plants is wrong written after: .... 6000 plnats .... .

Responses: Thank you very much. According to the referee’s suggestions, we have revised the details in lines 166 in the text.

In line 109: clarify what do you mean with the sentence “weights were recorded once they become constant”, a brief explanation to clarify this sentence is necessary.

Responses: Thank you very much. This sentence describes a method for determining the stable weight of a corn plant. Specifically, the plant is dried in an oven at 85°C for 48 hours, after which its weight is recorded once it has reached a stable level.

I consider that in the figures the conventional acronym of Mg (in capitalized letter M) was misused since it is assigned to magnesium. It seems that in true the legend of vertical axis should be say milligram; whose acronym is with lower letters: mg. Thus I recommend to use mg instead Mg

 Responses: Thank you very much. Mg is often considered as the weight of one hectare object in many literatures, such as yield, plant dry matter, etc.  I admit that it is the same as the traditional Mg acronym, but in this study it should be clearly indicated to the reader as corn yield or plant dry matter weight.

Figure 4a) Is r or R? Moreover, I suggest to include a space between the number and the ns word.

Responses: Thank you very much. According to the referee’s suggestions, we have revised the details in Figure 4a.

I could not find the figure 5, after figure 4 was presented figure 6, by the way, this last figure was referred in line 181 but with a dot that is not necessary (Figure. 6a).

Responses: Thank you very much. According to the referee’s suggestions, we have revised the details in Figure 5 and Figure 6.

In Line 183, was presented an interesting result .....plant density of 72,500 plants ha-1: Clarify, if this interesting result is that showed in the Figure 6a? If, it is, please refer the figure in this sentence.

Responses: Thank you very much. This result is shown in Figure 5a.

In the figure 7, it was confused that the ordinate has a negative value. Please give a brief explanation about this negative value.

Responses: Thank you very much. The ordinate does not have negative values in Figure 6.

This manuscript is a resubmission of an earlier submission. The following is a list of the peer review reports and author responses from that submission.

Round 1

Reviewer 1 Report

1. Lines 62 to 63, “However, the frequent occurrence of extreme weather has led to a huge loss of maize yield in recent years. Therefore,……” It felt like we were going to study the effects of extreme weather on corn, so let's rephrase that.

2. Lines 73 through 74 have incomplete parentheses.

3. What's 17.6 on line 80? available nitrogen?

4. Line 85 is 2020 or 2010?

5. Line Row 95 acres replaced with ha.

6. Line 99, subscript the 5 of O5.

7. Line 109, ℃ is the wrong label.

8. Line 131, Incomplete parenthesis.

9. Line 164-165, pre-silking DM and post-silking are not clear.

10. There is no description pre and post silking in 1.2, so I don't know how the dry matter of silking, pre- silking and post- silking in 3.1 was obtained.

11. The period of Line 243 is incorrect.

12. Line 192 ”Correspondongly” is wrong.

13. Inconsistent capitalization of titles in references.

Reviewer 2 Report

Dear Authors,

The subject of the influence of planting density and maize hybrids used in the manuscript is important for improving the yields obtained and thus ensuring food security.

I have some important comments on the content of the manuscript

I propose to include "maize hybrids" in the subject line

The introduction of the manuscript in my opinion needs some additions:

Line 47 State the average yields in the US. How does this relate to other countries in the world (Europe, Africa)? Why was the US specifically included? If there is a reason for this, it should be clearly stated

Line 59 - 62 What is the literature basis for this data

Line 62 - 63 What do the authors mean by extreme weather? Excessive precipitation? Drought? High low temperatures? Or other crop-damaging weather events?

Materials and Methods section:

Line 78 Evaluate whether these conditions were favorable to maize cultivation or rather unfavorable.

Line 81 Soil quality data are taken from the cited literature. In the cited manuscript, the experiment was conducted in 2012 and 2013. The experiment carried out by the authors in the extreme years differed by 7 years. Certainly, after such a period of time, the soil quality changed. It is essential to provide soil quality parameters in all years of the experiment at each location.

Line 85 2020?

Line 99 - 100 I suggest giving the amount of P and K in pure component and not in P2O5 and K2O, what type of fertilizers were used? How were weed diseases and pests controlled? This should be supplemented

Line 110 how was the harvesting done?

Line 120 - 121 previously mentioned the years 1990 - 2021

Results section

Line 147 Was there an interaction between years and hybrids or years and densities for grain yield? If so please describe it especially since in the introduction the authors mention losses due to weather conditions

Line 168 (table 3) Please add letters showing significant differences

Line 180 add reference to references section 

Line 190 add reference to references section

Line 191 This was also evaluated in the Authors' own research?

Discussion section

Line 237 If there was no radiation analysis in the Authors' own research, please indicate that this is based on results obtained by other authors

Line 257 It should be noted that this is the optimal planting density for the location where the study was conducted and the weather conditions from that area

Line 264 what stresses?

References section

Please standardize the names of publications, currently given in full names, abbreviations with and without periods.